# Examination of Green Productivity in China's Mining Industry: An In-Depth Exploration of the Role and Impact of Digital Economy

Chuandi Fang [1], Yue Yuan [2], Jiahao Chen [3], Da Gao [1,*] and Jing Peng [1]

[1] School of Law and Business, Wuhan Institute of Technology, Wuhan 430205, China; fangcd@wit.edu.cn (C.F.); pengjing@stu.wit.edu.cn (J.P.)
[2] School of Economics, Huazhong University of Science and Technology, Wuhan 430074, China; yyue@hust.edu.cn
[3] School of Economics and Management, China University of Geosciences, Wuhan 430078, China; chenjiahao_elevate@cug.edu.cn
[*] Correspondence: gaoda@hust.edu.cn

**Abstract:** Faced with the challenges of increasing demand and expanding emissions, China's mining industry is at a crucial stage of sustainable development. In the context of the new technological revolution and industrial transformation, researching how the digital economy can promote the growth of green total factor productivity (GTFP) in China's mining industry, particularly against the backdrop of technological diversity, is vital for achieving sustainable development and carbon neutrality goals. This study utilizes the meta-frontier Malmquist–Luenberger (MML) index to analyze the dynamics of GTFP in China's mining industry under technological heterogeneity. It thoroughly examines the direct and indirect impacts of the digital economy (DE) on GTFP and delves into the underlying mechanisms of these effects using the spatial Durbin model. The empirical results reveal a significant positive relationship between DE and GTFP, particularly pronounced in the areas of technical efficiency and technological catch-up. Notably, this study identifies the mediating role of industrial structural upgrading in linking DE and GTFP. Additionally, the observed spatial spillover effect of DE on local mining GTFP suggests that the influence of DE extends beyond the immediate regions within the mining sector. Based on these findings, the study presents policy recommendations, emphasizing the need to integrate cutting-edge digital technologies in mining to enhance environmental sustainability.

**Keywords:** digital economy; mining industry; green total factor productivity; technological heterogeneity; spatial spillover effect

## 1. Introduction

Amidst a profound transformation spanning key global sectors including economics, technology, culture, security, and politics, a new wave of technological revolution and industrial transformation is providing unprecedented development opportunities for nations around the globe. The Sustainable Development Goals (SDGs) set by the United Nations foreground the importance of harnessing such transformations to achieve a more inclusive, sustainable, and resilient future [1]. Within this transformative landscape, the digital economy has surfaced as a pivotal driver for global economic advancement, aligning with multiple SDGs by fostering innovation, ensuring equitable access to technological benefits [2], and promoting sustainable industrialization [3]. It enhances energy efficiency, contributes to Affordable and Clean Energy (SDG 7), promotes inclusive economic growth and job creation in line with Decent Work and Economic Growth (SDG 8), fosters sustainable industrialization and innovation, which is central to Industry, Innovation, and Infrastructure (SDG 9), and supports Climate Action (SDG 13) through technologies vital

for climate change mitigation. According to a 2022 white paper by the China Academy of Information and Communications Technology [4], the digital economy of 47 major global economies attained a remarkable $38.1 trillion in 2021, marking a year-over-year surge of $5.1 trillion (Figure 1). These statistics not only chart the ascendant course of the digital economy but also emphasize its deep-seated influence in domains such as semiconductors, artificial intelligence, digital infrastructure, and e-commerce and blockchain technology. Considering these developments, it is crucial to evaluate how the integration of digital economy strategies can further propel the realization of the SDGs and ensure a sustainable future for all.

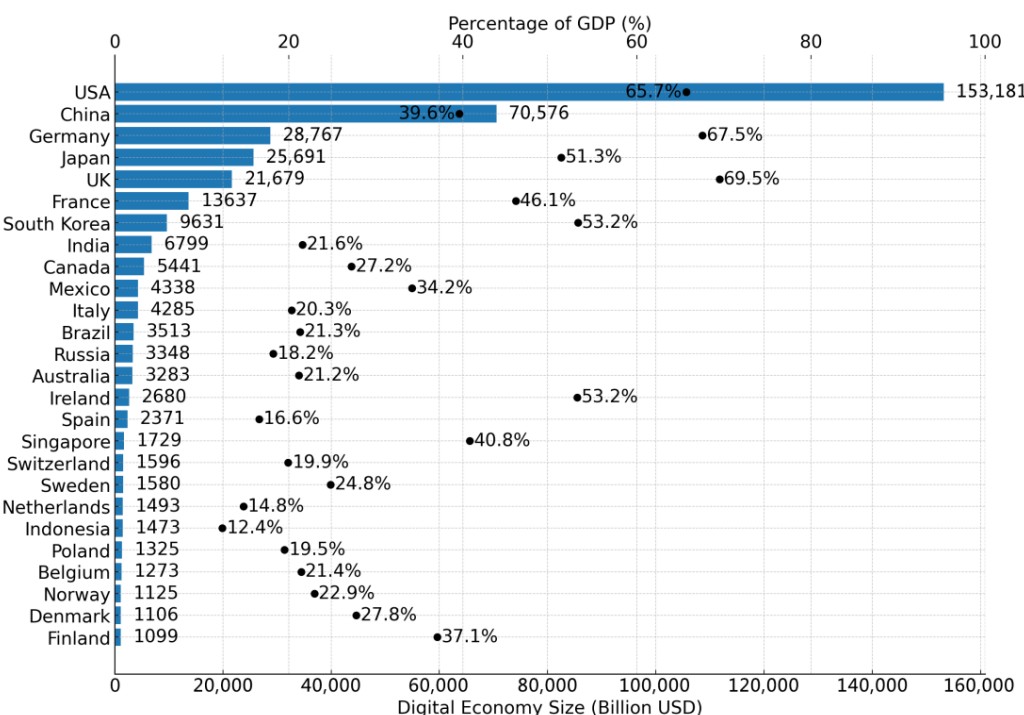

**Figure 1.** Digital economy size and its share of GDP in various countries, 2021. Data source: China Academy of Information and Communications Technology, 2022 [4].

The Chinese mining industry is facing challenges in sustainable development due to the dual expansion of supply demand and emissions. Data from the Annual Report 2022 for Carbon Dioxide Emission Accounts of Global Emerging Economies [5], for the year 2022 reveal that China's carbon emissions reached as high as 11 billion tons, accounting for 28.87% of global emissions. Industrial emissions, including those from mining, totaled 4.2 billion tons, comprising 38.18% of the country's overall emissions. These figures bring to light the mining industry's significant environmental responsibilities. Conversely, according to a report by the International Energy Agency [6], global energy demand is expected to grow annually by approximately 0.8% by 2030, primarily fueled by renewable and clean energy sources. This transition is likely to substantially increase the demand for essential mineral resources such as rare earth elements, lithium, and cobalt.

Therefore, it is particularly crucial to delve into how the digital economy can become a key driver in steering the mining industry towards a more green and efficient development path. Green Total Factor Productivity (GTFP), serving as a comprehensive tool for evaluating and promoting sustainable development, not only encapsulates the levels of resource efficiency and environmental protection, but also provides a framework for assessing technological innovation and ongoing transformation. Although digital technologies have been extensively researched and applied in tertiary industries such as services and finance, their specific roles and potential in the mining sector remain largely unexplored. Meanwhile,

existing research on GTFP seldom considers technological heterogeneity, making it an area that warrants focused attention.

This study aims to explore several key questions: First, what is the impact of the digital economy on GTFP in China's mining industry under technological heterogeneity? Second, how, and through which specific mechanisms, is the impact of the digital economy on green productivity in China's mining industry achieved? Finally, does this impact show significant regional or industrial differences, or is there a spatial spillover effect? Answering these questions will help deepen our understanding of the relationship between the digital economy and sustainable development and provide valuable insights and guidance for policymakers and industry stakeholders.

This study has the following marginal contributions. First, we use the meta-frontier Malmquist–Luenberger (MML) index to measure the GTFP of China's mining industry under technological heterogeneity. Secondly, to deeply explore the multi-dimensional impact of the digital economy on the heterogeneity GTFP of China's mining industry, we further decomposed it into technology progress index (TECH), technology efficiency change (EFCH), pure technology catch-up (PTCU) and potential technology relative change (PTRC). Finally, we also use the intermediary effect and the spatial spillover effect model to systematically explain how the digital economy concretely affects these key factors.

The remaining sections of this paper are arranged as follows: Section 2 briefly reviews the literature related to the DE and GTFP in the mining industry. Section 3 delves into a detailed mechanism analysis. Section 4 provides the methods and data used in the study. Section 5 presents all the empirical findings. Finally, Section 6 concludes and offers policy recommendations.

## 2. Literature Review

### 2.1. GTFP of the Mining Industry

GTFP has emerged as an indispensable tool for assessing economic efficiency while considering resource utilization and environmental impact. Originating from Robert Solow's [7] concept of Total factor productivity (TFP), it is evident that economic metrics needed to evolve, especially when Färe et al. [8] highlighted the potential shortcomings of TFP due to its omission of pollution as a detrimental output. This prompted scholars to integrate environmental and energy consumption factors, giving rise to the contemporary GTFP concept [9,10]. Methodologies to measure GTFP, particularly within China's mining landscape, have favored the Malmquist and the Malmquist–Luenberger (ML) indices [11]. Their applications have revealed profound insights: technological advancements predominantly bolster open-pit mining productivity [12], environmental regulations can hinder technological progress [13], and certain regions with ample investment see rapid GTFP growth in coal mining compared to metal mining [14].

Rambaldi et al. [15] define a meta-frontier function with a distance function and simultaneously construct a meta-frontier Malmquist productivity index and its decomposition using the Data Envelopment Analysis (DEA) method. Further expanding this field, Oh and Lee [16] introduced the MML index, a tool designed to account for efficiency heterogeneity among different groups. This index has shown promising results in addressing various aspects such as environmental efficiency [17], environmental productivity [18], carbon emission efficiency, and GTFP [19]. In contrast to the model proposed by Meeusen and van Den Broeck [20], which decomposes the ML indices into only technological and scale changes, the MML provides a more nuanced decomposition. It not only incorporates technological and scale changes but also differentiates between catch-up and potential relative technological changes [21]. This advanced decomposition methodology offers richer perspectives and frameworks for studying economic efficiency in depth. Currently, studies employing MML to analyze GTFP in the Chinese mining industry have demonstrated that foreign direct investment, environmental regulations, and innovation have a significant influence on GTFP [22].

### 2.2. Digital Economy

The meteoric rise of Information and Communication Technology (ICT) has positioned the digital economy at the epicenter of global economic and societal transformation. The digital economy, broadly defined as an economy energized by digital tools and data interchange [23], spans diverse sectors, from e-commerce and AI to IoT [24]. While universally accepted metrics for its measurement remain elusive, two predominant evaluation methodologies have emerged. The first emphasizes multidimensional frameworks such as the EU's Digital Economy and Society Index (DESI) and the World Economic Forum's Network Readiness Index (NRI). These frameworks aim to address the shortcomings of early singular metrics, such as mere internet penetration rates [25]. The second strategy focuses on the value added of the digital economy, splitting the analysis between industrial digitalization, which encompasses sectors like electronic device manufacturing [26], and digital industrialization, which tracks the value augmentation of traditional sectors via digital adoption [27]. From an economic perspective, the digital economy's impact is profound. On the micro level, it not only streamlines information flow but also spurs corporate innovation and heightens operational efficiency [28]. It also reshapes consumer behaviors by offering tailored and efficient services. On the macro scale, the digital economy recalibrates industrial dynamics, shifting from labor-centric models to tech-centric ones, accelerating economic growth, reinforcing industrial synergy, laying the groundwork for innovative infrastructure, and emphasizing sustainable solutions [29]. Recent research highlights the significant environmental impact of Industry 4.0 technologies, particularly on $CO_2$ emissions, underscoring the need for eco-conscious strategies in the digital economy's growth [30].

### 2.3. Digital Economy Effect on GTFP in the Mining Industry

The digital economy, with its transformative power, has emerged as a crucial catalyst for boosting GTFP, especially in the mining industry where renewable energy and digital innovation converge. Despite abundant evidence of the digital economy's beneficial impacts on diverse sectors such as enterprise management and manufacturing [31], scholarly exploration of its effects on mining productivity is scant. Current research indicates that melding the mining domain with state-of-the-art ICT offers a pathway to redefine its operational life cycle [32]. Innovations like big data analytics, AI, IoT, and cloud computing are poised not just to amplify exploration accuracy and extraction efficiency but also to make strides in environmental preservation and workforce safety [33]. Moreover, digital interventions are steering mining towards greener methodologies [34]. This metamorphosis involves integrated energy systems, automation, and ecological restoration, fortifying the sector's sustainable growth. Embracing these cutting-edge tools revitalizes the industry, laying the groundwork for its modernization and innovation [35].

To our knowledge, few studies have comprehensively analyzed the impact of the digital economy on the GTFP of China's mining industry from the perspective of technological heterogeneity. The marginal contribution of this paper lies in its innovative analysis of the impact of the digital economy on China's mining industry's GTFP from the perspective of technological heterogeneity, utilizing the MML index. Additionally, this study comprehensively explores the direct, indirect, and spatial impact mechanisms of the digital economy on GTFP in this sector through empirical analysis.

## 3. Mechanism Analysis

### 3.1. The Direct Effect of the Digital Economy on Mining GTFP

The digital economy's onset brings about pronounced direct influences on mining GTFP. Foremost, the progression of big data and machine learning offers mining companies unparalleled precision in predicting market trends and resource needs [36]. This precision reshapes production plans and slashes inventory expenses, directly boosting GTFP through optimal capital and labor deployment. In the environmental and risk oversight spheres, innovative tools like IoT and AI facilitate continuous monitoring, curtailing ecological

disasters and minimizing compliance perils [37]. This not only safeguards the environment but also strengthens GTFP. In optimizing supply chains and processes, the synthesis of cloud computing and real-time analytics grants mining entities unmatched oversight and adaptability [38]. This heightened management, coupled with sustainable production methods like real-time consumption tracking, paves the way for circular economy frameworks, propelling GTFP upward.

### 3.2. The Indirect Effect of the Digital Economy on Mining GTFP

The digital economy's growth imparts indirect influences on mining GTFP, primarily through the reshaping of industrial structures. Advanced technologies like artificial intelligence, big data analytics, IoT, and notably blockchain are redefining the mining sector, driving a pivot from resource-intensive methods to technology-driven, value-rich operations [39]. This restructuring amplifies the fusion of capital, labor, and tech resources, enhancing mining firms' global competitiveness and indirectly elevating GTFP. This revamped structure fosters the adoption of cutting-edge managerial strategies. Tools like intelligent analytics bolster mineral estimation precision, spurring operational efficiency [40]. Concurrently, digital training platforms enrich workforce expertise, contributing tangibly to GTFP. Blockchain technology enhances production efficiency and profitability by ensuring real-time transparency and immutability of supply chain data, indirectly improving financing efficiency, optimizing supply chain management, and reducing ineffective investments [41]. Amid rising global environmental concerns, the restructured industry leans towards sustainable practices. Technologies like smart sensors facilitate empirical evaluations of ecological footprints, offering avenues for sustainable production adaptations. Such evaluations reinforce corporate responsibility, solidifying the groundwork for GTFP's sustained augmentation.

### 3.3. Spatial Effect of the Digital Economy on Mining GTFP

The digital economy's effect on mining GTFP permeates beyond localized sectors, presenting pronounced externalities across vast regions. Central to this is the interplay of geographic agglomeration and network effects. Propelled by the digital economy, certain regions magnetize advanced mining tech, elite skills, and knowledge, elevating their GTFP [42]. This magnetism intensifies with network effects: as more entities adopt congruent tech or platforms, the network's collective value swells, fostering a smoother information flow and GTFP enhancement. Additionally, the digital medium catalyzes knowledge spillover and broadens innovation partnerships beyond regional confines [43]. Cutting-edge mining methodologies and managerial best practices disseminate swiftly, spurring wide-scale collaborations and enriching GTFP. The prowess of the digital economy, harnessed through data analytics and smart algorithms, refines supply chain and resource logistics to an unprecedented scale [44]. Its ripple effect is not restricted to certain pockets; it reverberates globally. This leads to precise supply chain fine-tuning, curbed production expenses, and efficient resource usage, amplifying global GTFP. Finally, the digital ecosystem is colored by regional policy nuances. Areas with robust digital policies and regulatory frames have a higher propensity to attract investments and skilled manpower, paving the way for GTFP disparities.

Building on the preceding mechanistic analyses (Figure 2), this paper posits the following hypotheses:

**Hypothesis 1:** *The digital economy exerts a positive impact on the GTFP within the mining industry.*

**Hypothesis 2:** *The digital economy indirectly elevates the GTFP in mining by facilitating the upgrading of industrial structures.*

**Hypothesis 3:** *There exists a significant geographical spillover effect of the digital economy, which further intensifies its influence on mining GTFP.*

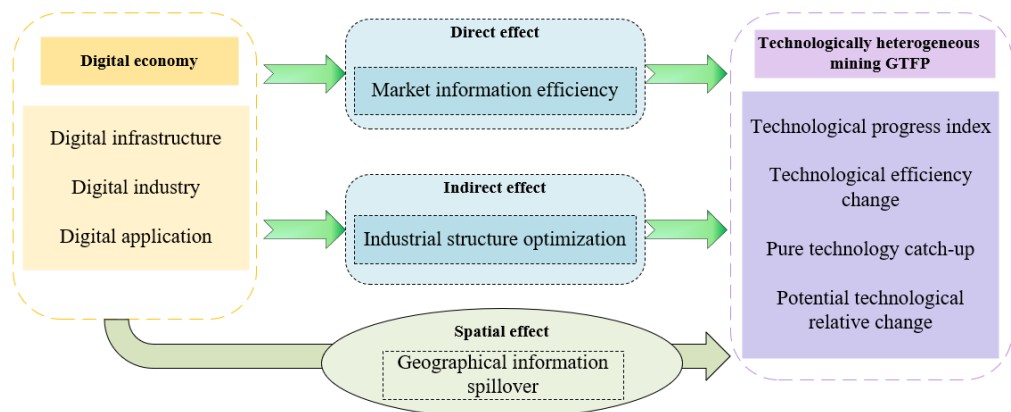

**Figure 2.** Mechanism analysis of digital economy on mining GTFP.

## 4. Methodology

### 4.1. Econometric Methods

#### 4.1.1. Baseline Model

In accordance with the studies of Ren et al. [45], we empirically examine the impact of the digital economy on the GTFP within the mining sector, using data from 30 provinces in China for the period spanning 2008–2021. The model is specified as in Equation (1):

$$GTFP_{it} = \alpha_0 + \alpha_1 DE_{it} + \alpha_2 DE^2{}_{it} + \alpha_3 Control_{it} + \lambda_i + \theta_t + \epsilon_{it} \tag{1}$$

where $GTFP_{it}$ represents the green total factor productivity in the mining industry, $DE$ stands for the digital economy, and $DE^2$ is the squared term of the digital economy introduced to capture potential nonlinear relationships between the digital economy and GTFP. The control variables are denoted by $Control_{it}$, $\theta_t$ is the time-fixed effect, $\lambda_i$ is the individual fixed effect, and $\epsilon_{it}$ is a stochastic error term.

In this study, we determined the statistical significance of coefficients using *p*-values, with thresholds set at less than 0.10, 0.05, and 0.01, corresponding to significance levels of 10%, 5%, and 1%, respectively. These levels are standard in econometric analyses and are chosen to balance the detection of true effects. Specifically, a 1% level indicates strong evidence against the null hypothesis, minimizing the likelihood of false significance claims, while a 5% level is widely used in social sciences for a moderate balance in findings. The 10% level is particularly useful in identifying subtle but notable trends in exploratory research or complex models where data limitations might reduce statistical power. *t*-tests provided t-values and corresponding *p*-values to assess each coefficient's statistical significance, and standard errors associated with each coefficient were used to determine the precision of estimation. For the estimation, we employ both Fixed Effects (FE) and Random Effects (RE) models and conduct a Hausman test to ascertain which model better fits the data.

To enhance the robustness of our analysis, the baseline model incorporates the following five control variables, which are consistently used in all subsequent models: (1) environmental regulation (ER), quantified by the ratio of completed investments in industrial pollution control to industrial added value. (2) Government intervention level (GIL), represented by the ratio of fiscal expenditure to GDP. (3) Energy structure (ES), expressed by the percentage of a region's electricity consumption relative to the national total. (4) Human capital level (HCL) quantified using the ratio of the number of students in higher education institutions to the total population. (5) Traditional postal services level (TPSL), measured by the total volume of postal and telecommunication services relative to GDP.

#### 4.1.2. Intermediary Effect Model

To delve deeper into the mechanisms through which the digital economy impacts the mining industry's GTFP, we incorporate the concept of mediation effects into our methodology. This approach allows us not only to deconstruct the underlying mechanisms but also to assess the relative contributions of multiple mediators, thus providing nuanced insights for sustainable development policies [46]. We employ the classic mediation effect model originally proposed by Baron and Kenny [47], with particular attention to the mediating role of industrial structure upgrading in transmitting the effects of the digital economy on mining GTFP. Based on Equation (1), we establish the following models:

$$M_{it} = \delta_0 + \delta_1 DE_{it} + \delta_2 DE^2_{it} + \delta_3 Control_{it} + \lambda_i + \theta_t + \epsilon_{it} \tag{2}$$

$$GTFP_{it} = \gamma_0 + \gamma_1 DE_{it} + \gamma_2 DE^2_{it} + \gamma_3 M_{it} + \gamma_4 Control_{it} + \lambda_i + \theta_t + \epsilon_{it} \tag{3}$$

#### 4.1.3. Spatial Durbin Model

This study aims to elucidate the relationship between the digital economy and GTFP in the mining sector, emphasizing the spatial associations often overlooked in previous research. Findings from past studies, such as those by Gu et al. [48], have demonstrated strong spatial correlations in the digital economy, underscoring the importance of spatial effects in understanding its impact on mining GTFP. To delineate this relationship more precisely, we employed Moran's I index to assess the spatial correlation of the digital economy and utilized both distance-based and 0–1 matrices to capture spatial relationships between cities. The expression for the Moran's I index is as follows:

$$Moran's\ I = \frac{\sum_{i=1}^{n} \sum_{j=1}^{n} W_{ij}(x_i - \overline{x})(x_j - \overline{x})}{S^2 \sum_{i=1}^{n} \sum_{j=1}^{n} W_{ij}} \tag{4}$$

The spatial Durbin model (SDM) was chosen to analyze the spatial spillover effects of the digital economy on mining GTFP. Through the SDM, this research provides insightful revelations into the intricate interplay between the digital economy and mining GTFP, particularly shedding light on spatial implications. The specific model is outlined as:

$$GTFP_{it} = \alpha_0 + \rho_1 \sum_{j=1}^{N} W_{ijt} GTFP_{it} + \beta_1 DE_{it} + \beta_2 DE^2_{it} + \beta_3 Control_{it} + \\ \rho_2 \sum_{i \neq j}^{N} W_{ijt} DE_{it} + \rho_3 \sum_{i \neq j}^{N} W_{ijt} DE^2_{it} + \rho_4 \sum_{i \neq j}^{N} W_{ijt} Control_{it} + \mu_i + \theta_t + \epsilon_{it} \tag{5}$$

For ease of subsequent analysis, the spatial weight matrix $W$ is calculated using geographic distance weights. $\mu_i$ and $\theta_t$ denote individual and time-fixed effects, respectively, while other variables are defined as in previous models.

#### 4.2. Variable Selection

#### 4.2.1. Digital Economy

This study investigates the relationship between the Digital Economy (DE) and the GTFP in China's mining industry. Given the absence of a unified DE evaluation criterion, the research refers to the G20 Hangzhou Summit's definition and indicators from the Digital Economy Development Report [49]. To enhance the analysis, this study has built an inclusive evaluation system focusing on three core dimensions: digital infrastructure, digital industries, and digital applications. Key indicators include Internet and mobile phone penetration rates, employment in Internet-related sectors, and an inclusive digital finance index. The entropy method standardizes this data, which has been used to compute the digital economy for Chinese provinces from 2008 to 2021. Details can be found in Table 1, with 2021's geographic trends showcased in Figure 3.

**Table 1.** Indicators for assessing digital economy development.

| Indicator Type | Indicator | Description and Unit |
| --- | --- | --- |
| Digital infrastructure | Internet penetration | Internet users per one hundred people |
| | Avg. telecom expenditure | Average telecom spending in yuan |
| | Mobile phone penetration | Mobile phone users per one hundred people |
| | Mobile base stations | Number of mobile communication base stations |
| Digital industry | Internet-related employees | Employees in computer and software (% of total) |
| | Digital industry scale | Revenue from digital tech (% of regional GDP) |
| | E-info investment | Fixed assets investment in billions of yuan |
| Digital application | Industrial IT level | Computers per one hundred employees in industrial enterprises |
| | DFI Index (PKU-DFIIC) | Peking University Digital Financial Inclusion Index |
| | E-commerce level | E-commerce sales (% of GDP) |

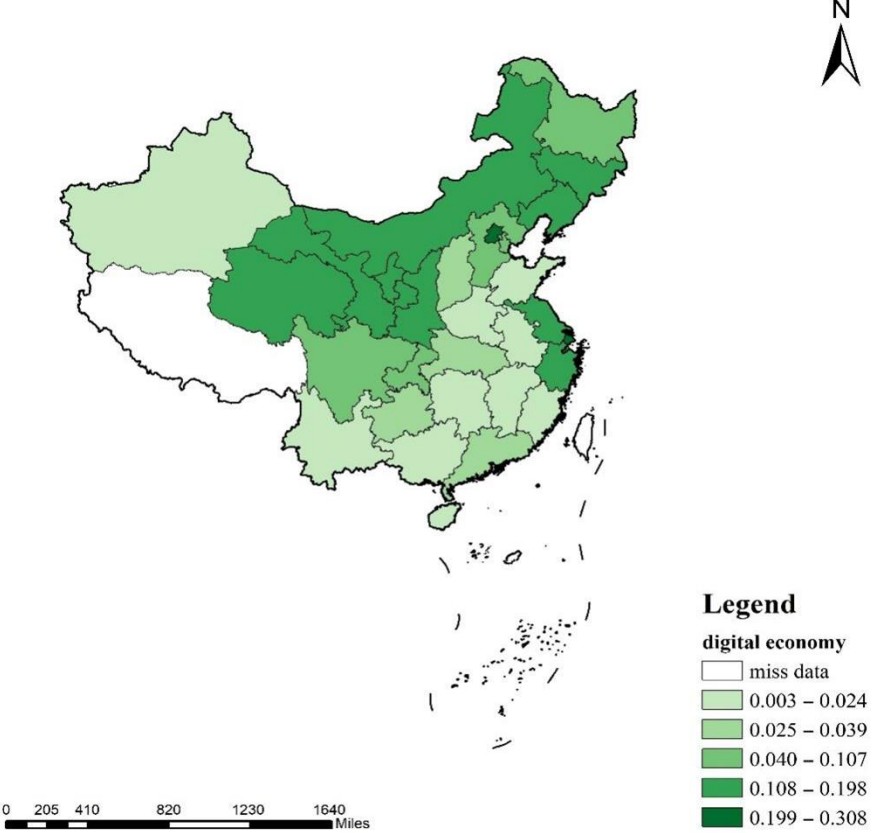

**Figure 3.** Geographic trends of the digital economy in 2021.

### 4.2.2. GTFP under Technological Heterogeneity

When considering the various stages of mining, traditional calculations of GTFP may be biased. Technological heterogeneity in the mining industry encompasses a wide range of factors, such as resource reserves, quality, storage conditions, spatial distribution, national strategies, and environmental impacts [50], all of which add complexity to Decision-Making Units (DMUs). Generally, geological conditions in adjacent areas tend to be similar, leading to comparable production technologies. Moreover, due to geographical barriers, more pronounced technological heterogeneity often emerges in remote areas, potentially leading to unique technological advancements [51].

To address this issue, Hayami [52] introduced the meta-frontier concept, which better represents technological diversity by dividing DMUs into specific subgroups [53]. This approach effectively addresses the technological variances in mining due to factors such as resource quality and environmental considerations. It begins by assessing the efficiency of each group-specific frontier, followed by establishing a collective meta-frontier. The

Technology Gap Ratio (TGR) is then used to measure the technological disparity between each subgroup and the meta-frontier, indicating the proximity to potential technological advancements. The GTFP in China's mining industry for the year 2021 is depicted in Figure 4. Define the directional vector as:

$$\vec{D}_c(x,y,c,g_y,-g_c) = max\{\lambda : (y + \lambda g_y, c - \lambda g_c) \in P(x)\} \tag{6}$$

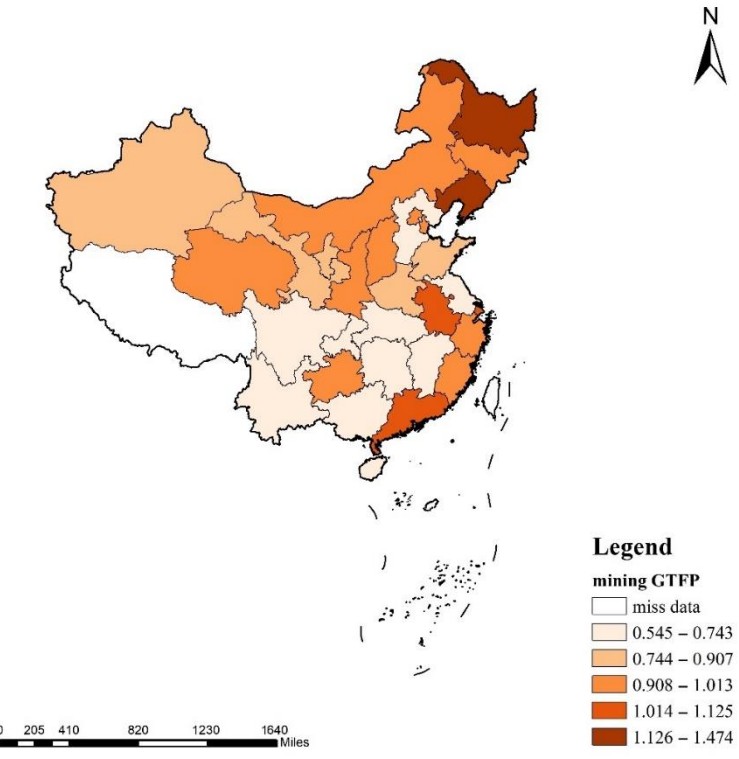

**Figure 4.** Geographic trends of GTFP in China's mining industry for the year 2021.

The notation $P(x)$ represents the production possibility set for mining inputs and outputs. The metrics for these inputs and outputs are in Table 2. Regarding the foundational form of the SBM-DEA model:

$$\vec{D}_c(x,y,c,g_y,-g_c) = min\rho = \frac{1 - \frac{1}{N}\sum_{n=1}^{N} S_n^x}{1 + \frac{1}{M+J}\left(\frac{\sum_{m=1}^{M} S_m^y}{y_{m0}} + \frac{\sum_{j=1}^{J} S_j^c}{c_{j0}}\right)} \tag{7}$$

$$s.t. \begin{cases} \sum_{i=1}^{I} z_i x_{i,n} - S_n^x = x_{i',n} \ n = 1, 2, \cdots N \\ \sum_{i=1}^{I} z_i y_{i,m} - S_m^y = y_{i',m} \ m = 1, 2, \cdots M \\ \sum_{i=1}^{I} z_i c_{i,j} - S_j^c = b_{i',j} \ j = 1, 2, \cdots J \\ S_n^x \geq 0, S_m^y \geq 0, S_i^b \geq 0, i = 1, 2, \cdots I \end{cases}$$

Moreover, we employ the *MML* index as an evaluative measure for productivity, capitalizing on both the meta-frontier and the group frontiers to aggregate efficiency information across diverse DMUs. Specifically, under the assumption of Variable Returns to Scale (VRS), the *MML* index can be decomposed into *TECH* and *EFCH*.

$$MML_{t+1}^t = \begin{bmatrix} \frac{1 + D_M^{t+1}(x^t, y^t, b^t; -x^t, y^t, -b^t)}{1 + D_M^{t+1}(x^{t+1}, y^{t+1}, b^{t+1}; -x^{t+1}, y^{t+1}, -b^{t+1})} \times \\ \frac{1 + D_M^t(x^t, y^t, b^t; -x^t, y^t, -b^t)}{1 + D_M^t(x^{t+1}, y^{t+1}, b^{t+1}; -x^{t+1}, y^{t+1}, -b^{t+1})} \end{bmatrix} \tag{8}$$

$$MML_{t+1}^{t} = TECH^M \times EFCH^M \tag{9}$$

$$EFCH^M = \frac{1 + D_M^t\left(x^t, y^t, b^t; -x^t, y^t, -b^t\right)}{1 + D_M^{t+1}\left(x^{t+1}, y^{t+1}, b^{t+1}; -x^{t+1}, y^{t+1}, -b^{t+1}\right)} \tag{10}$$

$$TECH^M = \begin{bmatrix} \frac{1 + D_M^{t+1}\left(x^t, y^t, b^t; -x^t, y^t, -b^t\right)}{1 + D_M^t\left(x^{t+1}, y^{t+1}, b^{t+1}; -x^{t+1}, y^{t+1}, -b^{t+1}\right)} \times \\ \frac{1 + D_M^{t+1}\left(x^t, y^t, b^t; -x^t, y^t, -b^t\right)}{1 + D_M^t\left(x^{t+1}, y^{t+1}, b^{t+1}; -x^{t+1}, y^{t+1}, -b^{t+1}\right)} \end{bmatrix} \tag{11}$$

Building on this, we introduce a new parameter, the technology adjustment factor (TAF), capturing the shift needed between technological levels and can be broken down into *PTCU* and *PTRC*.

$$TGR = \frac{1 + D_G^t\left(x^t, y^t, b^t; -x^t, y^t, -b^t\right)}{1 + D_M^t\left(x^{t+1}, y^{t+1}, b^{t+1}; -x^{t+1}, y^{t+1}, -b^{t+1}\right)} \tag{12}$$

$$TAF = \frac{MML}{GML} = \frac{TECH^M \times EFCH^M}{TECH^G \times EFCH^G} = \left[\frac{TGR_{t+1}(x_{t+1}, y_{t+1}, b_{t+1})}{TGR_t(x_t, y_t, b_t)} \times \frac{TGR_t(x_t, y_t, b_t)}{TGR_{t+1}(x_{t+1}, y_{t+1}, b_{t+1})}\right]^{\frac{1}{2}} \tag{13}$$

$$\begin{aligned} TAF &= \frac{TGR_{t+1}(x_{t+1}, y_{t+1}, b_{t+1})}{TGR_t(x_t, y_t, b_t)} \times \left[\frac{TGR_t(x_t, y_t, b_t)}{TGR_t(x_{t+1}, y_{t+1}, b_{t+1})} \times \frac{TGR_t(x_{t+1}, y_{t+1}, b_{t+1})}{TGR_{t+1}(x_t, y_t, b_t)}\right]^{\frac{1}{2}} \\ &= PTCU \times PTRC, \left(PTRC = \frac{TECH_{t,t+1}^M}{TECH_{t,t+1}^G}\right) \end{aligned} \tag{14}$$

The *MML* index can be decomposed into four components.

$$\begin{aligned} MML &= TECH^G \times EFCH^G \times \frac{TECH^M \times EFCH^M}{TECH^G \times EFCH^G} \\ &= TECH^G \times EFCH^G \times PTCU \times PTRC \end{aligned} \tag{15}$$

**Table 2.** Indicators for assessing mining GTFP.

| Indicator Type | Indicator | Description and Unit |
|---|---|---|
| | Energy consumption | Total energy consumed in mining (10,000 tons) |
| Input | Employment | Year-end employment in mining (people) |
| | Capital investment | New fixed-asset investment in mining (billion yuan) |
| | Water use | Total water used in mining (billion tons) |
| Expected output | Mining output value | Value of mining output (billion yuan) |
| | $SO_2$ emission | $SO_2$ emission in mining (tons) |
| Unintended output | Wastewater emission | wastewater emissions in mining (10,000 tons) |
| | Solid waste emission | Solid waste emissions in mining (tons) |

### 4.2.3. Mediating Variables

To comprehensively elucidate the intricate relationship between the DE and the GTFP in the mining sector, this study employs industrial structure upgrading (ISU) as a mediating variable. This index serves not only as an indicator of the level of economic development but also reveals potential associations with both the DE and mining sector GTFP. Building upon existing research [54], we have made appropriate modifications to the formula expressing ISU, which is as follows:

$$ISU_{it} = \sum_{j=1}^{3} g_{it} \times j \tag{16}$$

In this equation, $ISU_{it}$ represents the index of industrial structure upgrading for the *i* province in year *t*, with a range of $1 \leq ISU \leq 3$. The variable $g_{it}$ indicates the proportion of each sector in the total output value of the mining industry for the *i* province in year *t*.

The values of *j* are 1, 2, and 3, which correspond to the primary, secondary, and tertiary sectors, respectively.

### 4.3. Data Source

For the empirical analysis, we assembled a panel dataset spanning the years 2012 to 2021, which includes data from thirty provinces in China, encompassing municipalities and autonomous regions. Due to constraints related to the availability of reliable and comprehensive data, Tibet, Hong Kong, Macau, and Taiwan were not included in our sample. The primary data sources for this study comprise an array of authoritative publications, including the China Statistical Yearbook, China Mining Statistical Yearbook, China Land and Resources Statistical Yearbook, China Industrial Economic Statistical Yearbook, China Environmental Statistical Yearbook, and China High-tech Industry Statistical Yearbook. In addition, we utilized provincial statistical yearbooks and statistical data provided by the China Nonferrous Metals Industry Association. To maintain the integrity of our dataset, missing values were imputed using linear interpolation techniques. Detailed sample statistics can be found in Table 3.

**Table 3.** Descriptive statistics of the variables.

| Variable | Definition | Obs | Mean | Standard Deviation | Min | Max |
|----------|-----------|-----|------|--------------------|-----|-----|
| GTFP | Green total factor productivity | 420 | 1.009 | 0.11 | 0.545 | 1.584 |
| EFCH | Technological progress index | 420 | 1.012 | 0.219 | 0.297 | 2.511 |
| TECH | Technological efficiency change | 420 | 1.02 | 0.122 | 0.631 | 1.835 |
| PTCU | Pure technology catch-up | 420 | 1.044 | 0.41 | 0.261 | 5.405 |
| PTRC | Potential technological relative change | 420 | 1.009 | 0.164 | 0.435 | 1.89 |
| DE | Digital economy | 420 | 0.148 | 0.152 | 0.001 | 0.819 |
| ISU | Industrial structure upgrading | 420 | 2.379 | 0.127 | 2.132 | 2.834 |
| ER | Environmental regulation | 420 | 0.004 | 0.003 | 0 | 0.031 |
| GIL | Government intervention level | 420 | 0.239 | 0.1 | 0.087 | 0.643 |
| ES | Energy structure | 420 | 0.02 | 0.006 | 0.007 | 0.042 |
| HCL | Human capital level | 420 | 0.033 | 0.023 | 0.004 | 0.102 |
| TPSL | Traditional postal service level | 420 | 0.064 | 0.05 | 0.014 | 0.29 |

## 5. Results

### 5.1. Results on Mining GTFP

From 2008 to 2021, China's mining GTFP showed discernible regional variations. Overall, there was a slight upward trend. The Southwest, despite its −1.24% growth rate, maintained a commendable GTFP of 1.024. Conversely, the Northeast flourished with a GTFP of 1.016 and a growth rate of 2.11%. The North region presented complexities, balancing a GTFP of 1.020 with a −0.61% growth. The Eastern and Central regions faced challenges with negative growths, hinting at efficiency drops, especially in Central. At the provincial spectrum, Inner Mongolia led with a GTFP of 1.046 and a 3.43% growth. Anhui and Fujian followed suit, showcasing significant efficiencies. Conversely, Jiangsu and Hunan struggled with growth rates of −2.8% and −3.65%, pointing towards operational challenges. Examining GTFP's components, Central stood out in TECH with 1.039, trailed by Eastern's 1.032. However, the Southwest and Liaoning lagged. In EFCH, Guizhou led, whereas Hainan and Jiangsu were at the lower spectrum. Guizhou also excelled in PTCU, but Jiangsu and Hunan trailed. In PTRC, Hebei was prominent, while Guizhou and Liaoning showcased areas for enhancement. Figure 5 shows the average of mining GTFP from 2008 to 2021.

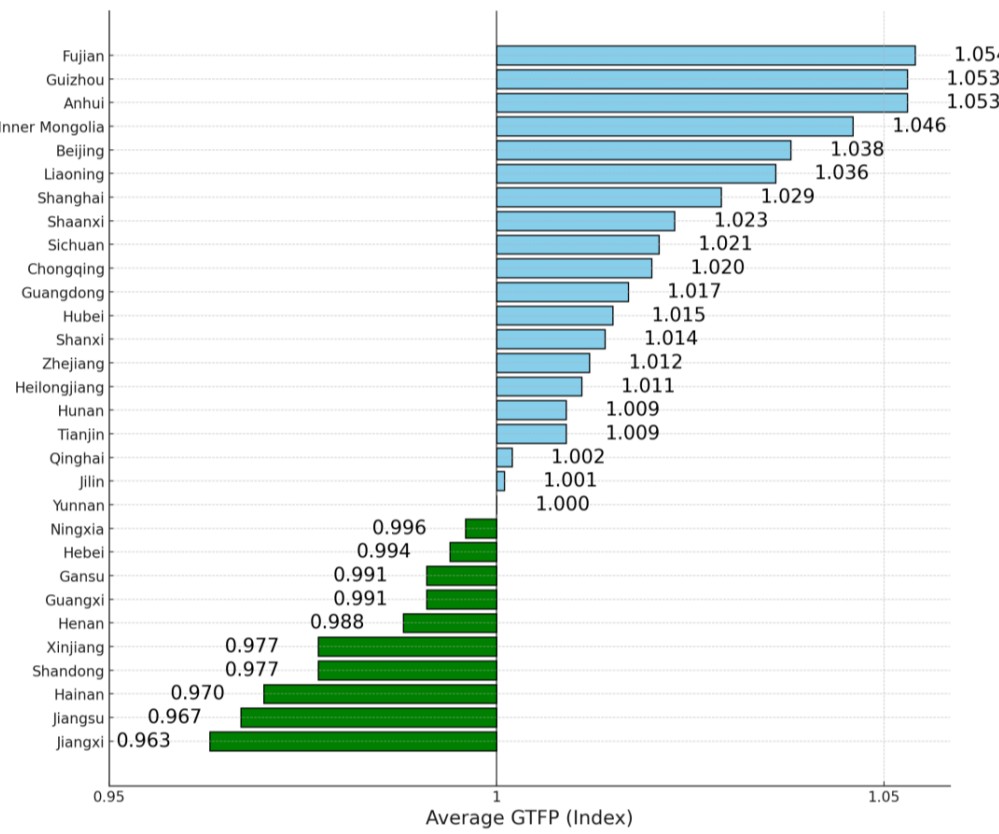

**Figure 5.** Average of mining GTFP (2008–2021).

*5.2. Results from the Baseline Regression Model*

5.2.1. Results

Using panel data from 30 Chinese provinces between 2008 and 2021, the relationship between the DE and mining GTFP was analyzed through Fixed Effects (FE) and Random Effects (RE) models. Table 4 shows the baseline regression results. The findings reveal that DE has a significant positive impact on GTFP, as evidenced by coefficients of 0.306 and 0.268, both significant at the 1% level. However, this influence is nonlinear, as indicated by the negative coefficients for the squared term of DE. The FE model, bolstered by the Hausman test, highlights an inverted U-shaped relationship between DE and GTFP. In the initial stages of the digital economy, technological innovation and increased efficiency significantly boosted the GTFP in the mining industry, leading to positive impacts. This was primarily due to the reduction in production costs and improvement in productivity brought about by early technological advancements. However, as the digital economy continues to evolve, these initial positive effects begin to diminish and may even turn negative. This shift could be attributed to the initial efficiency gains stimulating increased production activities, but over time, this might lead to excessive exploitation of resources and increased environmental pollution, adversely impacting GTFP.

It is important to note that the squared term of the digital economy only passed the statistical significance test at the 10% level under the FE model, whereas it was not statistically significant under the RE model. This indicates that while there are signs of an inverted U-shaped relationship, the relationship is not very clear or strong. This relatively marginal significance suggests that the impact of the digital economy on mining GTFP could be subject to a variety of complex factors, such as the pace and nature of technological change, market demand conditions, environmental policies, and specific industry environments. Therefore, in interpreting the long-term impact of the digital economy on mining GTFP, it is essential to consider these complex market dynamics and macroeconomic trends.

**Table 4.** Baseline regression results.

| Variables | GTFP | | EFCH | | TECH | | PTCU | | PTRC | |
|---|---|---|---|---|---|---|---|---|---|---|
| | FE | RE | FE | RE | FE | RE | FE | RE | FE | RE |
| DE | 0.306 *** | 0.268 *** | 0.596 *** | 0.579 *** | −0.375 *** | −0.327 *** | 0.971 ** | 1.023 *** | −0.396 *** | −0.406 *** |
| | (0.10) | (0.08) | (0.20) | (0.17) | (0.11) | (0.09) | (0.38) | (0.32) | (0.14) | (0.13) |
| DE$^2$ | −0.311 * | −0.218 | −0.668 * | −0.633 ** | 0.436 ** | 0.362 ** | −1.081 | −1.246 ** | 0.439 * | 0.440 ** |
| | (0.17) | (0.14) | (0.35) | (0.27) | (0.19) | (0.15) | (0.66) | (0.52) | (0.25) | (0.20) |
| ER | 2.21 | 2.65 | 3.18 | 4.33 | 2.44 | −3.977 ** | 8.17 | 5.06 | −6.199 * | −7.214 *** |
| | (2.19) | (1.69) | (4.45) | (3.39) | (2.40) | (1.87) | (8.47) | (6.46) | (3.23) | (2.51) |
| GIL | 0.26 | −0.197 *** | 0.51 | −0.342 ** | −0.363 * | 0.160 * | 0.64 | −0.154 | −0.577 ** | 0.16 |
| | (0.18) | (0.08) | (0.36) | (0.15) | (0.20) | (0.08) | (0.69) | (0.29) | (0.26) | (0.11) |
| ES | −10.19 *** | −3.533 *** | −18.23 *** | −7.369 *** | 14.44 *** | 4.465 *** | −12.66 | −7.726 * | 15.47 *** | 4.069 ** |
| | (2.02) | (1.07) | (4.12) | (2.16) | (2.22) | (1.19) | (7.84) | (4.14) | (2.99) | (1.60) |
| HCL | −1.082 | −0.633 ** | −3.347 | −1.395 ** | −0.793 | 0.764 ** | −3.448 | −1.22 | 0.54 | 0.832 * |
| | (1.69) | (0.30) | (3.44) | (0.60) | (1.86) | (0.33) | (6.56) | (1.16) | (2.50) | (0.45) |
| TPSL | 0.287 ** | 0.210 * | 0.415 * | 0.29 | −0.422 *** | −0.312 ** | 0.30 | 0.29 | −0.495 *** | −0.405 ** |
| | (0.12) | (0.11) | (0.24) | (0.23) | (0.13) | (0.12) | (0.45) | (0.43) | (0.17) | (0.17) |
| Constant | 1.126 *** | 1.094 *** | 1.265 *** | 1.194 *** | 0.921 *** | 0.935 *** | 1.111 *** | 1.142 *** | 0.917 *** | 0.955 *** |
| | (0.07) | (0.04) | (0.15) | (0.08) | (0.08) | (0.04) | (0.28) | (0.16) | (0.11) | (0.06) |
| Hausman | 23.81 | | 16.74 | | 16.74 | | | 4.15 | 14.74 | |
| | [0.001] | | [0.019] | | [0.000] | | | [0.761] | [0.039] | |
| R-squared | 0.12 | | 0.096 | | 0.158 | | 0.04 | | 0.125 | |
| Observations | 420 | 420 | 420 | 420 | 420 | 420 | 420 | 420 | 420 | 420 |
| Number of provinces | 30 | 30 | 30 | 30 | 30 | 30 | 30 | 30 | 30 | 30 |

Note: Standard errors are in parentheses. *** $p < 0.01$, ** $p < 0.05$, * $p < 0.1$.

This shift may be because although the digital economy initially promoted GTFP growth by improving EFCH and reducing costs, this growth has weakened over time. Specifically, the positive impact of the digital economy on EFCH, reflected by coefficients of 0.596 and 0.579, reveals the significant contribution of initial digitalization in enhancing efficiency and saving costs. However, in the realm of TECH, the digital economy has a negative impact, with coefficients of −0.375 and −0.327. This might reflect the ease of optimizing existing technological pathways as opposed to achieving deep technological innovations and breakthroughs. Despite the positive effect of the digital economy on PTCU, evidenced by coefficients of 0.971 and 1.023, demonstrating its capacity to push the industry towards or even meet current technological benchmarks, its effectiveness in the PTRC remains uncertain, with coefficients of −0.396 and −0.406. This suggests that while the digital economy has facilitated technological growth and catch-up, it may not be sufficiently effective in narrowing the gaps in technological innovation, leading to a gradual decline or shift towards a negative impact on GTFP in the long term.

5.2.2. Robustness Test Results

To ensure the reliability and accuracy of our research findings, we conducted a series of robustness tests on the baseline model. Initially, we adjusted the sample study period, selecting the time frame from 2012 to 2021 as the new research sample, and are presented in Table 5. The choice of this time range is inherently justified, as it coincides with the primary phase of 4G mobile network development in China, a period also characterized by rapid advancements in the DE. In the results obtained after this adjustment, the coefficients of the DE in the FE and RE models were 0.299 and 0.351, respectively. Both coefficients were statistically significant at a 5% level, reinforcing our belief that the DE has a sustained and significant positive impact on GTFP.

Subsequently, to further validate our models, we excluded data that included municipalities directly under the central government. As shown in Table 6, the coefficients for the DE in the FE and RE models were 0.254 and 0.307, respectively. The former was statistically significant at a 10% level, while the latter was significant at a 5% level. These findings are highly consistent with our baseline model, thereby further affirming the positive role of the DE in enhancing GTFP. The inconsistency in the significance of the squared term of DE across two robustness tests suggests that the trend of the inverted U-shaped curve is indeed not robust.

**Table 5.** Robustness test with adjusted sample study period.

| Variables | GTFP | | EFCH | | TECH | | PTCU | | PTRC | |
|---|---|---|---|---|---|---|---|---|---|---|
| | FE | RE | FE | RE | FE | RE | FE | RE | FE | RE |
| DE | 0.299 ** | 0.351 *** | 0.611 ** | 0.727 *** | −0.356 ** | −0.440 *** | 1.407 ** | 1.431 *** | −0.437 * | −0.589 *** |
| | (0.15) | (0.11) | (0.29) | (0.21) | (0.17) | (0.12) | (0.62) | (0.45) | (0.23) | (0.17) |
| DE² | −0.317 | −0.336 ** | −0.789 | −0.873 *** | 0.47 | 0.529 *** | −1.732 | −1.820 *** | 0.56 | 0.697 *** |
| | (0.25) | (0.17) | (0.50) | (0.32) | (0.29) | (0.19) | (1.07) | (0.70) | (0.39) | (0.26) |
| ER | 3.38 | 3.443 * | 6.11 | 6.215 * | −3.993 | −5.602 ** | 12.16 | 5.08 | −8.355 ** | −9.172 *** |
| | (2.59) | (1.95) | (5.07) | (3.78) | (2.94) | (2.25) | (10.95) | (8.20) | (4.02) | (3.06) |
| GIL | 0.38 | −0.141 | 0.886 * | −0.14 | −0.313 | 0.10 | 1.956 * | 0.15 | −0.637 | 0.03 |
| | (0.27) | (0.10) | (0.53) | (0.20) | (0.31) | (0.12) | (1.14) | (0.43) | (0.42) | (0.16) |
| ES | −11.74 *** | −3.645 ** | −18.51 *** | −5.635 * | 18.29 *** | 3.969 ** | −7.198 | −5.789 | 18.53 *** | 2.29 |
| | (3.16) | (1.53) | (6.19) | (2.95) | (3.59) | (1.76) | (13.37) | (6.46) | (4.91) | (2.39) |
| HCL | −1.432 | −0.441 | −3.368 | −0.69 | −0.0326 | 0.61 | 0.73 | 0.29 | −0.129 | 0.10 |
| | (2.84) | (0.40) | (5.57) | (0.78) | (3.23) | (0.47) | (12.04) | (1.72) | (4.42) | (0.63) |
| TPSL | 0.311 ** | 0.19 | 0.35 | 0.20 | −0.534 *** | −0.288 * | −0.177 | 0.02 | −0.544 ** | −0.297 |
| | (0.15) | (0.13) | (0.29) | (0.25) | (0.17) | (0.15) | (0.63) | (0.53) | (0.23) | (0.20) |
| Constant | 1.133 *** | 1.061 *** | 1.165 *** | 1.060 *** | 0.813 *** | 0.985 *** | 0.50 | 0.952 *** | 0.902 *** | 1.069 *** |
| | (0.15) | (0.06) | (0.29) | (0.12) | (0.17) | (0.07) | (0.62) | (0.26) | (0.23) | (0.10) |
| Hausman | 21.69 | | 13.19 | | 29.28 | | 4.81 | | 11.92 | |
| | [0.000] | | [0.070] | | [0.000] | | [0.680] | | [0.100] | |
| R-squared | 0.143 | | 0.114 | | 0.186 | | 0.052 | | 0.146 | |
| Observations | 300 | 300 | 300 | 300 | 300 | 300 | 300 | 300 | 300 | 300 |
| Number of provinces | 30 | 30 | 30 | 30 | 30 | 30 | 30 | 30 | 30 | 30 |

Note: Standard errors are in parentheses. *** $p < 0.01$, ** $p < 0.05$, * $p < 0.1$.

**Table 6.** Robustness test excluding direct-administered municipalities.

| Variables | GTFP | | EFCH | | TECH | | PTCU | | PTRC | |
|---|---|---|---|---|---|---|---|---|---|---|
| | FE | RE | FE | RE | FE | RE | FE | RE | FE | RE |
| DE | 0.254 * | 0.307 ** | 0.515 * | 0.633 *** | −0.294 ** | −0.331 ** | 0.934 * | 1.258 *** | −0.229 | −0.410 ** |
| | (0.13) | (0.12) | (0.27) | (0.24) | (0.15) | (0.13) | (0.52) | (0.46) | (0.20) | (0.18) |
| DE² | −0.065 | −0.316 | −0.224 | −0.719 | 0.09 | 0.36 | −0.647 | −1.815 * | −0.158 | 0.46 |
| | (0.30) | (0.27) | (0.62) | (0.54) | (0.33) | (0.30) | (1.18) | (1.03) | (0.44) | (0.40) |
| ER | 2.14 | 2.41 | 3.73 | 4.16 | −3.146 | −3.776 * | 9.36 | 5.43 | −7.097 ** | −7.362 *** |
| | (2.35) | (1.84) | (4.81) | (3.72) | (2.57) | (2.03) | (9.20) | (7.07) | (3.46) | (2.73) |
| GIL | 0.22 | −0.199 ** | 0.43 | −0.360 ** | −0.334 | 0.169 * | 0.40 | −0.195 | −0.548 * | 0.17 |
| | (0.20) | (0.08) | (0.41) | (0.17) | (0.22) | (0.09) | (0.78) | (0.31) | (0.29) | (0.12) |
| ES | −9.817 *** | −4.528 *** | −17.42 *** | −8.527 *** | 14.12 *** | 5.915 *** | −9.996 | −8.409 | 15.00 *** | 5.682 *** |
| | (2.29) | (1.42) | (4.68) | (2.88) | (2.50) | (1.57) | (8.95) | (5.47) | (3.37) | (2.11) |
| HCL | −1.507 | −0.587 * | −4.127 | −1.387 ** | −0.38 | 0.709 * | −3.875 | −1.309 | 1.10 | 0.72 |
| | (1.84) | (0.33) | (3.76) | (0.66) | (2.01) | (0.36) | (7.18) | (1.26) | (2.70) | (0.49) |
| TPSL | 0.281 ** | 0.219 * | 0.42 | 0.31 | −0.436 *** | −0.346 ** | 0.28 | 0.28 | −0.518 *** | −0.452 ** |
| | (0.13) | (0.12) | (0.26) | (0.25) | (0.14) | (0.13) | (0.49) | (0.47) | (0.18) | (0.18) |
| Constant | 1.131 *** | 1.107 *** | 1.277 *** | 1.213 *** | 0.929 *** | 0.912 *** | 1.118 *** | 1.152 *** | 0.922 *** | 0.934 *** |
| | (0.08) | (0.05) | (0.17) | (0.09) | (0.09) | (0.05) | (0.32) | (0.18) | (0.12) | (0.07) |
| Hausman | 19.59 | | 14.73 | | 30.55 | | 6.53 | | 22.61 | |
| | [0.010] | | [0.040] | | [0.000] | | [0.480] | | [0.000] | |
| R-squared | 0.119 | | 0.097 | | 0.16 | | 0.041 | | 0.129 | |
| Observations | 364 | 364 | 364 | 364 | 364 | 364 | 364 | 364 | 364 | 364 |
| Number of provinces | 26 | 26 | 26 | 26 | 26 | 26 | 26 | 26 | 26 | 26 |

Note: Standard errors are in parentheses. *** $p < 0.01$, ** $p < 0.05$, * $p < 0.1$.

### 5.3. Mediating Effects

To delve deeper into the mechanisms by which the DE influences GTFP, we introduced industrial structure upgrading as a mediating variable. This decision stems from an understanding of the complex relationship between shifts in industrial structure and ecological sustainability in modern economic development. We utilized an index representing the sophistication of the industrial structure to operationalize this variable. The results are presented in Table 7. Initially, we found that the DE and ISU were significantly negatively correlated at the 5% significance level, with a coefficient of −0.263. This suggests that as the DE expands, traditional sectors may face pressure to downsize or transform, triggering a comprehensive adjustment of the industrial structure, especially those characterized by high energy consumption and emissions. Subsequently, we noted that the ISU was negatively correlated with GTFP at a 5% significance level, with a coefficient of −0.0947. While this may initially appear counterintuitive, it likely reflects the elevated capital, technological, and resource inputs required in the nascent stages of industrial upgrading. Such overinvestment could exacerbate environmental pressure in the short term, thereby temporarily reducing GTFP. When considering the combined influence of DE and ISU, we found that DE still had a statistically significant positive impact on GTFP at the 1% level, with a coefficient of 0.296. This reaffirms the digital economy's enduring and positive role in fostering green productivity.

**Table 7.** Regression results for the intermediate effects of industrial structure upgrading.

| Variables | GTFP | EFCH | TECH | PTCU | PTRC | ISU |
|---|---|---|---|---|---|---|
| DE | 0.296 *** | 0.573 *** | −0.364 *** | 0.953 ** | −0.383 *** | −0.263 ** |
| | (0.10) | (0.20) | (0.11) | (0.38) | (0.14) | (0.11) |
| DE$^2$ | −0.304 * | −0.651 * | 0.428 ** | −1.068 | 0.429 * | 0.30 |
| | (0.17) | (0.35) | (0.19) | (0.66) | (0.25) | (0.21) |
| ISU | −0.0947 ** | −0.227 *** | 0.107 ** | −0.175 | 0.132 ** | |
| | (0.04) | (0.09) | (0.05) | (0.17) | (0.06) | |
| ER | 1.87 | 2.37 | −2.055 | 7.55 | −5.727 * | −3.57 |
| | (2.18) | (4.43) | (2.39) | (8.49) | (3.22) | (2.59) |
| GIL | 0.19 | 0.34 | −0.285 | 0.52 | −0.481 * | −0.724 *** |
| | (0.18) | (0.36) | (0.20) | (0.70) | (0.27) | (0.21) |
| ES | −11.08 *** | −20.36 *** | 15.45 *** | −14.31 * | 16.71 *** | −9.407 *** |
| | (2.05) | (4.17) | (2.25) | (8.00) | (3.03) | (2.40) |
| HCL | −0.883 | −2.868 | −1.02 | −3.079 | 0.26 | 2.11 |
| | (1.69) | (3.42) | (1.85) | (6.57) | (2.49) | (2.01) |
| TPSL | 0.308 *** | 0.465 ** | −0.446 *** | 0.34 | −0.524 *** | 0.22 |
| | (0.12) | (0.23) | (0.13) | (0.45) | (0.17) | (0.14) |
| Constant | 1.379 *** | 1.873 *** | 0.634 *** | 1.580 *** | 0.564 *** | 2.677 *** |
| | (0.14) | (0.28) | (0.15) | (0.53) | (0.20) | (0.09) |
| R-squared | 0.13 | 0.11 | 0.17 | 0.04 | 0.14 | 0.11 |
| Observations | 420 | 420 | 420 | 420 | 420 | 420 |
| Number of provinces | 30 | 30 | 30 | 30 | 30 | 30 |

Note: Standard errors are in parentheses. *** $p < 0.01$, ** $p < 0.05$, * $p < 0.1$.

Further, we conducted a detailed analysis of the specific effects of the DE on the decomposed elements of GTFP. Specifically, DE had a significant positive impact on EFCH, verified at both 1% and 5% significance levels, with coefficients of 0.596 and 0.579, respectively. This likely indicates that the rise of DE has accelerated improvements in technical efficiency within the industrial sector, thus elevating the overall level of operational efficiency. However, the impact on TECH was negative, with coefficients of −0.375 and −0.327, both statistically significant at the 1% level. This could be attributed to the challenges in achieving deeper technological advancements, such as the difficulty in transitioning to new technological paradigms, high costs, and the need for extensive personnel training and cultural adaptation, which may temporarily inhibit advancements in technology. On the aspect of PTCU, the DE showed a positive impact, with coefficients of 0.971 and 1.023, significant at the 5% and 1% levels, respectively. This suggests that the DE helps firms approach or achieve the current technological frontier more quickly, thereby increasing overall productivity. Finally, regarding PTRC, the impact was negative, with coefficients of −0.396 and −0.406, both significant at the 1% level. This implies that while the DE aids in technological progress and pure technical catch-up, its role in narrowing the potential technical gap remains less evident.

*5.4. Spatial Effects*

5.4.1. Spatial Correlation Test

This study investigates the spatial correlation characteristics of DE. To accomplish this, we employed both adjacency weight matrices and geographic distance weight matrices to calculate Moran's I index. The results are presented in Table 8. In many of the years examined, Moran's I index showed positive values and was statistically significant at a 5% level or lower, irrespective of whether adjacency or geographic distance weight matrices were used. This finding strongly suggests a significant positive spatial autocorrelation among geographically adjacent regions in terms of digital economic activity. This result not only underscores the importance of geographic factors in the development of DE but also enhances the robustness of the conclusion.

**Table 8.** Moran's index is based on adjacency matrix and geographical distance matrix.

| Year | Adjacency | | Geographical Distance | |
|------|-----------|-----------|------------------------|-----------|
| | Z | *p*-Value | Z | *p*-Value |
| 2008 | 3.8723 | 0.0001 | 3.9480 | 0.0001 |
| 2009 | 3.5067 | 0.0005 | 3.4261 | 0.0006 |
| 2010 | 3.0384 | 0.0024 | 2.0744 | 0.038 |
| 2011 | 1.7049 | 0.0882 | 0.5901 | 0.5552 |
| 2012 | 3.3772 | 0.0007 | 3.8657 | 0.0001 |
| 2013 | 3.6149 | 0.0003 | 3.7184 | 0.0002 |
| 2014 | 4.3845 | 0.0000 | 4.9744 | 0.0000 |
| 2015 | 2.9135 | 0.0036 | 3.8501 | 0.0001 |
| 2016 | 3.3971 | 0.0007 | 3.4011 | 0.0007 |
| 2017 | 3.3848 | 0.0007 | 3.3398 | 0.0008 |
| 2018 | 4.1392 | 0.0000 | 4.0055 | 0.0001 |
| 2019 | 1.0821 | 0.2792 | 0.1452 | 0.8845 |
| 2020 | 0.9237 | 0.3556 | −0.2348 | 0.8144 |
| 2021 | 2.9095 | 0.0036 | 2.7315 | 0.0063 |

However, there were a few years where spatial correlation was not markedly evident. This could potentially be attributed to specific events or unaccounted-for variables during those periods, such as the COVID-19 pandemic that began in 2019. It is worth mentioning that while the global Moran's I index can somewhat homogenize inter-provincial disparities, it is not entirely capable of capturing the local spatial correlations within individual provinces. To address this, we specifically calculated local spatial correlations for the years 2015 and 2021, as depicted in Figure 6. The results further corroborated the presence of significant positive spatial correlations and clustering phenomena.

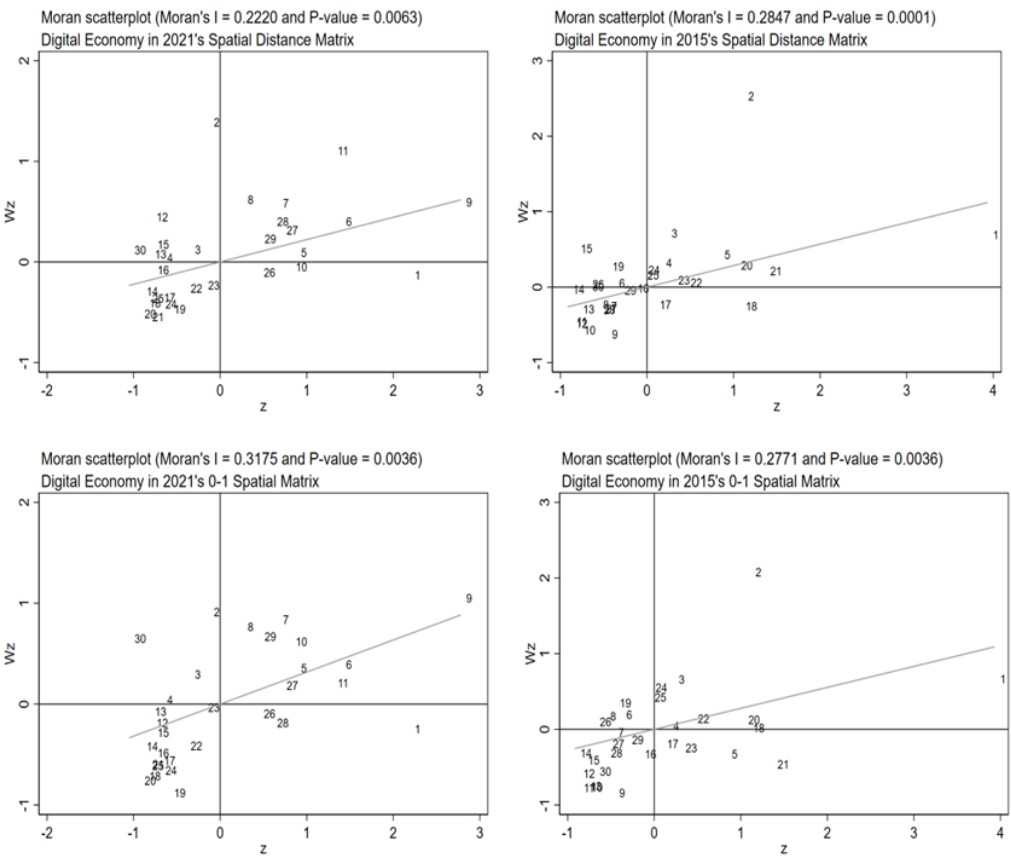

**Figure 6.** Scatter plot of local Moran's I in 2015 and 2021.

5.4.2. Analysis of Spatial Effects

Amidst the digitalization trend and Industry 4.0, DE has had profound implications across various sectors, most notably in mining. Our spatial analysis of the relationship between the DE and GTFP in the mining sector reveals noteworthy observations. The results are presented in Table 9. First, the direct influence of the DE on mining GTFP is positive. Specifically, the correlation coefficient between the DE and GTFP was measured at 0.260, statistically significant at a 5% level. This result unequivocally demonstrates the pivotal role that DE plays in enhancing both efficiency and sustainability in mining. This is attributed not only to digital technologies' ability to improve productivity and reduce costs but also to their capacity to facilitate more effective resource management and minimize environmental impact through data analytics and intelligent decision making. More strikingly, the DE not only influences local mining productivity but also manifests a strong spillover effect in the spatial dimension. This is evidenced by an indirect impact coefficient of 0.512, signifying that the development of the DE in one region not only uplifts its own mining GTFP but also positively affects adjacent regions. This territorial spillover is likely the result of various interacting factors, including but not limited to knowledge sharing, technology transfer, and capital mobility. However, it should be noted that we observed a certain negative influence of the square term of DE on GTFP. This observation may suggest that although the initial stages of DE implementation can significantly enhance mining GTFP, excessive digitalization may lead to diminishing returns. This is not to assert that further development of the DE will inevitably have negative consequences, but rather to emphasize the need for caution regarding potential risks and challenges such as data security, technology dependence, and social and environmental impacts.

**Table 9.** Regression results of SDM model.

| Variables | Direct Effect | | | | | Spillover Effect | | | | |
|---|---|---|---|---|---|---|---|---|---|---|
| | GTFP | EFCH | TECH | PTCU | PTRC | GTFP | EFCH | TECH | PTCU | PTRC |
| DE | 0.260 ** | 0.546 *** | −0.305 *** | 0.835 ** | −0.322 ** | 0.512 * | 0.85 | −0.659 ** | 1.584 | −0.65 |
| | (0.10) | (0.21) | (0.11) | (0.40) | (0.15) | (0.27) | (0.56) | (0.31) | (0.98) | (0.40) |
| $DE^2$ | −0.288 * | −0.642 * | 0.375 ** | −1.035 | 0.371 | −0.797 ** | −1.453 * | 1.011 ** | −2.477 * | 1.039 * |
| | (0.17) | (0.34) | (0.18) | (0.65) | (0.25) | (0.39) | (0.83) | (0.47) | (1.45) | (0.61) |
| ER | 2.041 | 3.008 | −1.331 | 8.341 | −4.671 | 0.436 | 0.784 | −0.561 | 1.226 | −1.635 |
| | (1.99) | (4.04) | (2.17) | (7.71) | (2.94) | (0.55) | (1.29) | (0.99) | (1.75) | (1.27) |
| GIL | 0.0639 | 0.18 | −0.0653 | 0.0665 | −0.254 | 0.0114 | 0.0429 | −0.0247 | 0.000791 | −0.0858 |
| | (0.19) | (0.38) | (0.20) | (0.72) | (0.27) | (0.05) | (0.11) | (0.09) | (0.13) | (0.11) |
| ES | −8.961 *** | −15.32 *** | 10.87 *** | −12.54 | 12.22 *** | −1.906 ** | −4.020 ** | 4.375 *** | −1.78 | 4.118 *** |
| | (2.10) | (4.19) | (2.34) | (7.66) | (3.06) | (0.95) | (1.91) | (1.34) | (1.82) | (1.49) |
| HCL | −1.216 | −3.463 | −0.503 | −3.608 | 0.891 | −0.269 | −0.942 | −0.205 | −0.539 | 0.32 |
| | (1.61) | (3.27) | (1.75) | (6.25) | (2.37) | (0.42) | (1.07) | (0.79) | (1.23) | (0.91) |
| TPSL | 0.195 * | 0.25 | −0.249 * | 0.09 | −0.305 * | 0.04 | 0.07 | −0.100 * | 0.01 | −0.103 |
| | (0.12) | (0.24) | (0.13) | (0.45) | (0.18) | (0.03) | (0.07) | (0.06) | (0.08) | (0.07) |
| rho | 0.179 ** | 0.212 *** | 0.297 *** | 0.148 *** | 0.259 *** | 0.179 ** | 0.212 *** | 0.297 *** | 0.148 *** | 0.259 *** |
| | (0.08) | (0.08) | (0.07) | (0.01) | (0.07) | (0.08) | (0.08) | (0.07) | (0.01) | (0.07) |
| Log-L | 375.7427 | 77.7747 | 342.6651 | −186.419 | 214.8062 | 375.7427 | 77.7747 | 342.6651 | −186.419 | 214.8062 |

Note: Standard errors are in parentheses. *** $p < 0.01$, ** $p < 0.05$, * $p < 0.1$.

Regarding the decomposed components of GTFP, our analysis revealed important insights. DE significantly and positively affects EFCH and PTCU, further affirming the digital economy's ability to enhance technical efficiency and advancements in mining technology. Conversely, TECH and PTRC were negatively impacted, possibly due to the increased complexity and uncertainty introduced by the widespread application of digital technologies. Overall, the DE exerts a highly positive effect on mining GTFP, with a total coefficient of 0.772, statistically significant at a 1% level. This finding not only reaffirms the centrality of the DE in augmenting mining productivity but also suggests that future policy formulation should consider the multi-dimensional effects and influences of the DE to achieve more sustainable and efficient mining operations.

This study examines the impact of the digital economy on the GTFP in the mining industry, employing a GTFP calculation method like that used in the manufacturing sector as per the literature [55]. We found that while the digital economy initially has a positive impact on mining, it may shift to an inverted U-shaped curve over time, in contrast to

the manufacturing sector where significant effects become apparent only after reaching a certain threshold of investment. Additionally, consistent with the literature [56], we identify industrial structure upgrading as crucial for efficiency and low-carbon development. We also observed spatial spillover effects of the digital economy, aligning with the findings of the literature [57,58], which highlight its extensive impact across different industries and regions.

## 6. Conclusions and Policy Implications

This study utilizes panel data from 30 provinces, municipalities, and autonomous regions in China, spanning from 2008 to 2021, to estimate the GTFP under technological heterogeneity in the mining sector through the MML index model. To delve further into the impact of the DE on mining GTFP, we also employed the spatial Durbin model and mediation effect models for empirical verification. The study yields the following conclusions: (1) The findings validate a positive influence of the DE on GTFP, which is statistically significant at the 1% level. While the DE contributes positively to technological progress and pure technical catch-up, its role in enhancing technical efficiency and reducing the potential technological gap remains inconclusive. Control variables such as environmental regulations and government intervention also have varying degrees of influence on GTFP and its components. (2) The study uncovers the mediating role of industrial structure upgrading in the relationship between the DE and GTFP. Specifically, the DE exerts a constrictive effect on industrial structure, which may yield negative repercussions for GTFP in the short term. Through component analysis, we further ascertain that the DE positively influences technological progress and pure technical catch-up. (3) Beyond the direct positive impact of the DE on local mining GTFP, our research identifies a significant spatial spillover effect. This implies that the benefits of the DE extend beyond elevating local mining productivity to affecting adjacent regions, thereby exhibiting its inter-regional diffusion and influence.

Based on the above conclusions, this paper derives the following policy implications.

(1) Promote and apply deep technological innovation. By providing research and development funding, tax incentives, and fostering public–private sector cooperation, real technological innovation, and breakthroughs are stimulated. Implementing technology demonstration projects, promoting industry best practices, and offering financial incentives support businesses in adopting and applying these innovative technologies.

(2) Industry transformation for environmental efficiency. Governmental strategies should focus on the connection between DE, GTFP, and industrial structural change. Publicly announced goals should emphasize green operations, low carbon emissions, and value addition. Financial incentives can guide polluting industries towards eco-friendliness. Environmental management should span the entire industrial process, ensuring genuine eco-friendly transformations. These measures will harmonize economic growth with environmental responsibilities, further promoting sustainable GTFP.

(3) Cross-regional collaboration for sustainable mining. Given the DE's impact on mining efficiency and its regional spillovers, a strategy promoting regional mining sustainability is vital. Establishing Centers for Digital Innovation in Mining can foster knowledge sharing and tech transfers. A cross-regional environmental regulatory body should ensure unified eco-standards and oversight. Financial incentives, like awards, can motivate companies to focus on regional environmental impacts. These recommendations push for leveraging DE's potential to amplify regional sustainability in mining, aligning individual mining progress with wider regional green objectives.

Considering the findings from our study, particularly the observed inverted U-shaped relationship between the DE and GTFP, future research directions should focus on deepening the understanding of this complex relationship. It is essential to explore the causes behind the dominant negative effects that emerged, especially in the context of the U-shaped curve we identified. Further investigation could benefit from employing heterogeneous panel estimators, which would allow for a more nuanced examination of the thresholds

within this relationship. Specifically, utilizing panel thresholds could provide valuable insights into the varying impacts of DE on GTFP across different regions or sectors.

**Author Contributions:** Conceptualization, C.F.; Methodology, C.F., Y.Y., J.C. and D.G.; Software, Y.Y. and J.C.; Validation, Y.Y. and J.C.; Formal analysis, J.C. and J.P.; Resources, D.G. and J.P.; Data curation, J.P.; Writing—original draft, C.F.; Writing—review & editing, C.F., Y.Y. and D.G.; Visualization, J.P.; Supervision, D.G.; Funding acquisition, Y.Y. All authors have read and agreed to the published version of the manuscript.

**Funding:** This research was funded by National Natural Science Foundation of China grant numbers 71991482, 72104189 and 72204235.

**Institutional Review Board Statement:** Not applicable.

**Informed Consent Statement:** Not applicable.

**Data Availability Statement:** The data presented in this study are available on request from the corresponding author.

**Conflicts of Interest:** The authors declare no conflicts of interest.

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
