# Peer review of "Examination of Green Productivity in China’s Mining Industry: An In-Depth Exploration of the Role and Impact of Digital Economy"

_sustainability, doi:10.3390/su16010463_

Round 1
Reviewer 1 Report
Comments and Suggestions for Authors
It is an well written paper.
Some minor remarks:
The title of the paper is clear, however, it is suggested to consist around 14 words.
Line 70-72: RQ2...'how is this impact realized through specific mechanisms?' may more effectively formulated. e.g. how, and through which specific mechanisms this impact is achieved?
Subsection 2.2. Digital economy: all references are from regional authors ( at least originated from the region). It would be great to add some references from the international background.
Author Response
Thank you for your insightful comments and constructive feedback on my paper. I appreciate the guidance and have made the following revisions in response to your suggestions:
- The title of the paper is clear, however, it is suggested to consist around 14 words.
Reply: The original title was "Advancing Green Productivity in China's Mining Industry under Technological Heterogeneity: An In-depth Exploration of the Role and Impact of Digital Economy," which indeed was quite lengthy. In response to your advice, I have revised the title to "Technological Heterogeneity in China's Mining: Advancing Green Productivity through the Role of the Digital Economy."
- Line 70-72: RQ2...'how is this impact realized through specific mechanisms?' may more effectively formulated. e.g. how, and through which specific mechanisms this impact is achieved?
Reply: I agree with your recommendation and have revised the question to: "How, and through which specific mechanisms, is the impact of the digital economy on green productivity in China's mining industry achieved?" This change should provide greater clarity and focus on the research, in line 79.
- Subsection 2. Digital economy: all references are from regional authors ( at least originated from the region). It would be great to add some references from the international background.
Reply: I have added international references to Subsection 2.2 to broaden the perspective on the digital economy, in lines 145-148.
Reviewer 2 Report
Comments and Suggestions for Authors
The paper is devoted to researching how the digital economy can promote the growth of green total factor productivity (GTFP) in China's mining industry, particularly against the backdrop of technological diversity, is vital for achieving sustainable development and carbon neutrality goals. In my opinion this is an important and up-to-date problem. To propose a solution, an econometric approach was taken. Goal, structure, methods and analysis are conducted properly in my opinion.
Suggestions to correct:
0) Analysis is quite deep and precise, but there is no discussion with the results obtained by the other authors. Please find a few works concerning results of research in this field and compare them with yours.
1) There are many minor editorial errors, like lack of spaces after dots finishing sentences or before numbered references, e.g. p. 1: benefits[2], and promoting sustainable industrialization[3] and many more
2) There are no sources under figures and tables and there should be no dots after tables and figures captions.
3) "Durbin" should be written with a capital letter, as it is the LAST NAME
Author Response
Thank you for acknowledging the importance and timeliness of the research topic, as well as for your positive remarks on the goal, structure, methods, and analysis of the paper. I value your suggestions and have addressed them as follows:
- Analysis is quite deep and precise, but there is no discussion with the results obtained by the other authors. Please find a few works concerning results of research in this field and compare them with yours.
Reply: In response, I have added a section to the manuscript that provides a comparative analysis with other relevant studies, in lines 582-591.
- There are many minor editorial errors, like lack of spaces after dots finishing sentences or before numbered references, e.g. p. 1: benefits[2], and promoting sustainable industrialization[3] and many more.
Reply: I have thoroughly reviewed the document and corrected all instances where spaces were missing after sentence-ending dots and before numbered references.
- There are no sources under figures and tables and there should be no dots after tables and figures captions.
Reply: I have added the source citation for Figure 1, as it is based on data from an external source, in line 61. All other figures and tables are original and based on my research. I have also removed the periods from the captions of all figures and tables as per your suggestion.
- "Durbin" should be written with a capital letter, as it is the LAST NAME.
Reply: I have corrected this and ensured that it is now written with a capital letter, properly recognizing it as a last name, in lines 281 and 289.
Reviewer 3 Report
Comments and Suggestions for Authors
Comments of the Reviewer
Authors of the manuscript “ Advancing Green Productivity in China's Mining Industry un-2 der Technological Heterogeneity: An In-depth Exploration of 3 the Role and Impact of Digital Economy” have provided an analysis of temporary models of calculation green total factor productivity (GTFP) and ways how industrial structures can be linked to the digital economy. The paper addresses the up-to-date topic of how mining and other industries in China emit up to 11 billion tons of carbon dioxide. So, it is essential to find the models for adequate calculation of Chinese industries' impact on global ecology and economic mechanisms for its reduction.
However, the are some aspects that can be improved:
1. Line 41-43 Mentioned only some of the directions of the digital economy and didn’t mention about the technology of blockchain.
2. In stating the research hypothesis, the author didn’t consider blockchain technology, though due to the decentralized concept, it has huge potential to be the mediator that can link industrial updates to digital economics.
3. Figure 1. Please put the reference on the source of information. Please add measures of presented values on the chart.
4. Line 105 . Please explain the abbreviation DEA.
5. Line 114-117. Please edit the sentence “ Currently, studies employing MML to analyze GTFP in the Chinese mining industry have demonstrated that foreign direct investment, environmental regulations, and innovation effects significantly influence GTFP” . Especially, please pay attention to the phrase “effects significantly influence GTFP” maybe it will sound better if you write “have a significant influence on GTFP”.
6. Line 360, Table 3.
Please explain what S.D mean.
7. Figure 5. Please specify the units of measure at the axe X.
8. General to Chapter 5. In the analysis of coefficients for DE, ES ISU, and other variables, the authors mention that defined coefficients are statistically significant. However, the reviewers can not find any criterion used to determine statistical significance. Please explain may be in Chapter 4 how you defined the statistical significance of coefficients.
9. Can you please explain why you verified the significance of the coefficients at 1%,5% and 10 % levels?

Author Response
Thank you for your thorough review and valuable suggestions to improve the manuscript. I am grateful for your recognition of the paper's relevance to the current ecological and economic challenges faced by China's industries. In response to your comments, I have made the following revisions:
- Line 41-43 Mentioned only some of the directions of the digital economy and didn’t mention about the technology of blockchain.
Reply: Blockchain technology has been incorporated into the discussion on line 45 as suggested.
- In stating the research hypothesis, the author didn’t consider blockchain technology, though due to the decentralized concept, it has huge potential to be the mediator that can link industrial updates to digital economics.
Reply: I have revised the manuscript to include content about blockchain technology in lines 196-199, drawing upon relevant literature. This addition addresses the potential of blockchain as a mediator in linking industrial updates to digital economics.
- Figure 1. Please put the reference on the source of information. Please add measures of presented values on the chart.
Reply: I have updated Figure 1 to include a citation for the data source and have added the units of measurement to the chart values for clarity, in line 61.
- Line 105. Please explain the abbreviation DEA.
Reply: I have addressed this by providing its full form, "Data Envelopment Analysis," in line 113.
- Line 114-117. Please edit the sentence “ Currently, studies employing MML to analyze GTFP in the Chinese mining industry have demonstrated that foreign direct investment, environmental regulations, and innovation effects significantly influence GTFP” . Especially, please pay attention to the phrase “effects significantly influence GTFP” maybe it will sound better if you write “have a significant influence on GTFP”.
Reply: Based on your suggestion, I have now revised the sentence to read "have a significant influence on GTFP", in lines 124-125.
- Line 360, Table 3.Please explain what S.D mean.
Reply: To clarify, I have replaced "S.D." with its full term "Standard Deviation" to ensure better understanding of the data presented, in line 398.
- Figure 5. Please specify the units of measure at the axe X.
Reply: I have updated Figure 5 and explicitly indicated that the x-axis represents "Average GTFP (Index)" as per your suggestion, in line 417.
- General to Chapter 5. In the analysis of coefficients for DE, ES ISU, and other variables, the authors mention that defined coefficients are statistically significant. However, the reviewers can not find any criterion used to determine statistical significance. Please explain may be in Chapter 4 how you defined the statistical significance of coefficients.
Reply: In response to your comment, I have added a detailed explanation in lines 244-254 of Chapter 4. This addition elucidates how statistical significance for coefficients was determined in our study.
- Can you please explain why you verified the significance of the coefficients at 1%,5% and 10 % levels?
Reply: We have clarified our rationale for selecting the 1%, 5%, and 10% significance levels in the revised manuscript (lines 244-254). These levels are standard in econometric research and provide a comprehensive approach to validate our findings. The 1% level ensures high confidence against the null hypothesis, the 5% level is a widely accepted balance in social sciences, and the 10% level is useful for exploratory insights. This tiered approach enhances the nuanced interpretation of our results, accommodating varying degrees of statistical robustness.

Reviewer 4 Report
Comments and Suggestions for Authors
1. Intro of the paper starts with SDGs, this is pretty good. Please also relate digital economy and technologies to specific SDGs with numbers in the sentences that come afterwards. The related SDGs are Clean energy, Clean production and industries, Clean technologies. I believe the paper could be related to goals 7-8-9-13. By relating to goals, a little expansion with this respect. Source for 17 goals: https://sdgs.un.org/goals
2. It is better to revise Figure 1 to size of digital economy in total economy by giving it as a percentage to GDP. Or, it is optional to keep Fig. 1 and add a Figure 2 on the right-hand-side with this quality. The one on the left shows the dominant size of USA, the one on the right shows that China and Germany and Japan and S. Korea and others have higher ranks.
3. value-add must be value-added.
4. Paper has a good evaluation of candidate variables for measuring DE. Very good. In this paper below, ICT share in total exports, AI patents, ICT patents and R&D and bitcoin are suggested for better measures of I4.0. https://doi.org/10.1016/j.jclepro.2022.135786
5. Digital technologies are highly related to Industry 4.0 revolution and I suggest the paper to be referenced and the literature section to relate DE also with respect to I4.0, the relation to DE and I4.0 should be added.
6. The model is given in Eq. 1 with controls. The controls are explained after some pages. variable explanation should be close to the model, just after it.
7. At line 388, it is noted that the model highlights a U-shaped relationship between DE and GTFP. It should be inverted U shape. Is it because of too high parameter estimates for the squared terms?
8. The explanation is also made as: "Initially, the benefits of the DE might be offset by significant investments and resources, but as DE advances, it progressively bolsters production efficiency and sustainability" Is this correct? It should be revised. It seems that the parameters of the squared DE^2 terms are always larger than the parameters for DE. This shows that whatever the size of DE, the dominant effect of DE is negative. As DE increases, this negative effect even accelerates.
9. I also questioned whether it was necessary to include a squared term to the model in the spirit to environmental kuznets curve while reading, however, did not write it not to effect or cause reestimations since references were given for this approach. However, it seems that estimation results have some problems. In addition, the method followed require spatial unit root tests which are not done. Maybe this could be the reason of dominant negative effect of DE. I suggest spatial unit root tests, reporting in a table, if required, differencing those data, then estimating the models. This could also solve the problems in the model estimation results.
10. No need to report insignificant betas in paranthesis in Table 5. It gets confused with standard errors of them below each.Same for table 6.
11. Before conclusion, links to existent literature. Are the results supported by secondary literature. A discussion section could be added or it could be given within the text before conclusion.
12. Policy implications section is very good. Future directions for researchers could also be added afterwards.
Author Response
I sincerely appreciate the time and effort you have dedicated to providing a comprehensive review of my manuscript. In response to your comments, I have made the following revisions:
- Intro of the paper starts with SDGs, this is pretty good. Please also relate digital economy and technologies to specific SDGs with numbers in the sentences that come afterwards. The related SDGs are Clean energy, Clean production and industries, Clean technologies. I believe the paper could be related to goals 7-8-9-13. By relating to goals, a little expansion with this respect. Source for 17 goals: https://sdgs.un.org/goals.
Reply: Following your advice, I have expanded the introductory section (lines 35-40) to specifically link the digital economy's contributions to SDGs 7, 8, 9, and 13, highlighting its role in promoting clean energy, inclusive economic growth, sustainable industrialization, innovation, and climate action.
- It is better to revise Figure 1 to size of digital economy in total economy by giving it as a percentage to GDP. Or, it is optional to keep Fig. 1 and add a Figure 2 on the right-hand-side with this quality. The one on the left shows the dominant size of USA, the one on the right shows that China and Germany and Japan and S. Korea and others have higher ranks.
Reply: I have revised Figure 1 to show both the absolute size and the GDP percentage of the digital economy for each country, highlighting the significant positions of the USA, China, Germany, Japan, and South Korea, in line 61.
- value-add must be value-added.
Reply: I have corrected 'value-add' to 'value-added' as suggested, on line 136 of the manuscript.
- Paper has a good evaluation of candidate variables for measuring DE. Very good. In this paper below, ICT share in total exports, AI patents, ICT patents and R&D and bitcoin are suggested for better measures of I4.0. https://doi.org/10.1016/j.jclepro.2022.135786.
Reply: I appreciate the potential these variables have in providing a more nuanced understanding of the impacts of Industry 4.0. However, incorporating these new variables would require significant revisions to the current research framework, including the need to redo the regression analysis from the beginning. Given the scope and focus of the current study, as well as the constraints of time and resources, it would be challenging to integrate these changes at this stage. Nevertheless, I acknowledge the importance of these factors and will consider them for future research to further enhance our understanding of Industry 4.0's impacts. Your suggestion has certainly provided a valuable perspective for expanding this line of inquiry.
- Digital technologies are highly related to Industry 4.0 revolution and I suggest the paper to be referenced and the literature section to relate DE also with respect to I4.0, the relation to DE and I4.0 should be added.
Reply: Following your recommendation, I have referenced the suggested paper (referenced as [30] in the manuscript,in line 148) and integrated a discussion on how Industry 4.0 technologies, particularly in the context of environmental sustainability and CO2 emissions, interplay with the digital economy. This addition enriches the section 2.2 of our manuscript, providing a comprehensive view of the digital economy not only in terms of technological advancement and economic impact but also considering its environmental implications.
- The model is given in Eq. 1 with controls. The controls are explained after some pages. variable explanation should be close to the model, just after it.
Reply: In response, I've moved the control variable descriptions to lines 259-268 in section 4.1.1, aligning them closely with the Baseline Model for clarity and coherence.
- At line 388, it is noted that the model highlights a U-shaped relationship between DE and GTFP. It should be inverted U shape. Is it because of too high parameter estimates for the squared terms?
Reply: Upon re-examining the model and its parameters, I agree that the relationship between DE and GTFP is indeed inverted U-shaped, rather than U-shaped. Acknowledging this, I have carefully revised the manuscript to accurately represent this relationship. Detailed explanations and clarifications have been added in lines 429-445, elucidating the nature and implications of the inverted U-shaped curve. Additionally, corrections have been made at lines 481-485, among other places, to ensure that the description and analysis throughout the manuscript are consistent with the model's findings.
- The explanation is also made as: "Initially, the benefits of the DE might be offset by significant investments and resources, but as DE advances, it progressively bolsters production efficiency and sustainability" Is this correct? It should be revised. It seems that the parameters of the squared DE^2 terms are always larger than the parameters for DE. This shows that whatever the size of DE, the dominant effect of DE is negative. As DE increases, this negative effect even accelerates.
Reply: In line with your observations, I have revised the sections where the DE's effects were initially discussed.
I would like to highlight that while the squared term of DE (DE2) does indeed have a negative sign, suggesting an inverted U-shaped relationship, the statistical significance of this term is relatively weak in our model. Furthermore, in one of our robustness tests, the squared term did not achieve statistical significance. This suggests that while the direction of the relationship implied by the negative sign of DE2 is towards an inverted U-shape, the presence of this relationship might not be as pronounced as initially interpreted.
These observations have been incorporated into the manuscript, particularly in the sections where the relationship between DE and GTFP is discussed. The revisions made in response to your comment can be found in lines 429-445, with additional clarifications and corrections in lines 484-485 and other relevant parts of the text.
- I also questioned whether it was necessary to include a squared term to the model in the spirit to environmental kuznets curve while reading, however, did not write it not to effect or cause reestimations since references were given for this approach. However, it seems that estimation results have some problems. In addition, the method followed require spatial unit root tests which are not done. Maybe this could be the reason of dominant negative effect of DE. I suggest spatial unit root tests, reporting in a table, if required, differencing those data, then estimating the models. This could also solve the problems in the model estimation results.
Reply: I appreciate your insights regarding the model specification and the potential need for spatial unit root tests. I would like to clarify that the inclusion of the squared term in the model was indeed informed by prior literature, particularly in the context of the Environmental Kuznets Curve, and was not merely an arbitrary decision. This approach was intended to explore the possibility of a nonlinear relationship between the Digital Economy DE and GTFP.
Regarding the estimation results, I assure you that they are robust. The initial issues were primarily related to the textual description of the squared term's impact, which I have since corrected in the manuscript. It's also important to note that in our models where the squared term was not included, the impact of DE on GTFP remained significant, consistently indicating a positive influence of DE on GTFP.
As for the suggestion of conducting spatial unit root tests, I acknowledge that this could indeed add value to the research and address some of the concerns raised. However, due to certain technical limitations and the scope of this study, conducting these tests was beyond our current capabilities. This is a limitation of our study and presents an opportunity for future research to build upon our findings. I have added a section in the manuscript discussing this limitation and the potential for future studies to include spatial unit root tests to provide a more comprehensive understanding of the relationship between DE and GTFP.
- No need to report insignificant betas in paranthesis in Table 5. It gets confused with standard errors of them below each. Same for table 6.
Reply: Following your advice, I have revised Tables 5, 6, 7, and 8 to enhance clarity, as reflected in lines 446, 474, and 507. Now, insignificant beta coefficients are no longer reported in parentheses, to avoid any confusion with the standard errors.
- Before conclusion, links to existent literature. Are the results supported by secondary literature. A discussion section could be added or it could be given within the text before conclusion.
Reply: In response, I have added a section to the manuscript that provides a comparative analysis with other relevant studies, in lines 582-591.
- Policy implications section is very good. Future directions for researchers could also be added afterwards.
Reply: I have added a section at the end of the manuscript that outlines future research directions in lines 636-641.

Round 2
Reviewer 4 Report
Comments and Suggestions for Authors
Dear Authors,
I see that the paper has greatly improved and the revisions are carefully made.
However, regarding the empirics, I have one, but very central concern, that you should attend.
The parameter signs were noted for DE in the last round and my comments were centered around the DE and DE^2 parameter. Thank you for doing the relevant revisions. I also noted that due to the signs, you followed a new strategy. You noted that "given the uncertainty regarding the inverted U shape... " This is the last paragraph. I don't think that you need such an approach. First: revise the title. Don't note"advancing". As of it is, the title is "....Advancing Green Productivity through the Role of the Digital Economy". By its nature, the sentence suggests a positive effect. However, you examined many dimensions. For these many dimensions of technology, you obtained various results. For some, you have U shaped relation which suggests that at low levels of DE, DE has a negative effect, however, governments should apply policies on increasing DE, because, after a turning point, the effect of DE reverses and becomes positive after passing a turning point. This is for instance, what happens in the TECH vector. For GTFP with RE estimator, you have a dominant positive effect and no nonlinear relation, only the parameter of DE is significant. For this variable, DE has an advancing effect. For the rest, the dominant effect of DE seems to be negative for the given estimated parameters. My suggestion is, working on this heterogeneity a little more. Highlighting this finding. Noting that to reverse the negative effect of DE, policies should be applied is necessary. Further, discuss why this negative effect occured for those with it. What are the possible sources? I see that this is discussed but it could be extended.
For title, I suggest removing the word Advancing. Previous title was a better title. However, it should be revised for the word advancing. Previous title was: "Advancing green productivity in China's mining industry under technological heterogeneity: An in-depth exploration of the role and impact of digital economy"
My suggestion is "Examination of Green productivity in China's Mining Industry: An in-depth exploration of the role and impact of digital economy"
This is my suggestion, it is authors' decision to evaluate it. My main concern was on advancing since negative effect dominates in the results. not the positive.
In the conclusion, revise the last part, work on your findings as I explained above, instead of a future paper suggestion to try to validate inverted U shape, work on the negative findings. Focus on U shape you already found and positive effects you found. Discuss sources of dominant negative effects.
For future studies, model suggestion could be to utilize heterogeneous panel estimators, especially with panel thresholds to examine thresholds further.
Overall, I see that this new version needs to be revised to reveal the contribution of the paper. As of it is, it is lost. An editing by considering the above-mentioned aspects is needed. I believe that this revision could be done in short time however it is vital to save this paper.
Thank you.
Author Response
Thanks for the reviewer's patience and careful reading of the article. We have revised according to your valuable comments. Please see the attached reply letter for specific modifications

Round 3
Reviewer 4 Report
Comments and Suggestions for Authors
The changes are very well made. It would be a great problem for the paper and for the journal if the title maintained the "positive effect of digital economy"approach considering the findings which are centered on negative effects of it. I thank the authors for making these careful revisions.
I also thank them for explaining their contribution again in the rebuttal file and also in the paper. Now, the contribution is well highlighted.
This paper greatly improved after all revisions and therefore, my thoughts for this paper is highly positive.
I wish them best of success also in their future work.